# Exo70 Promotes the Invasion of Pancreatic Cancer Cells via the Regulation of Exosomes

**DOI:** 10.3390/cancers16020336

**Published:** 2024-01-12

**Authors:** Jingzhou Xiang, Bowen Zheng, Lingying Zhao, Yuting He, Fanzhuoran Lou, Runyang Li, Miao Fu, Xintian Huang, Wenqing Zhang, Xiaoting Hong, Li Xiao, Tianhui Hu

**Affiliations:** 1Xiamen Key Laboratory for Tumor Metastasis, Cancer Research Center, School of Medicine, Xiamen University, Xiamen 361102, China; 24520220157343@stu.xmu.edu.cn (J.X.); 24520220157288@stu.xmu.edu.cn (B.Z.); zhaolingying@stu.xmu.edu.cn (L.Z.); 24520211154655@stu.xmu.edu.cn (R.L.); 24520231154559@stu.xmu.edu.cn (M.F.); wqzhang@xmu.edu.cn (W.Z.); xthong@xmu.edu.cn (X.H.); 2National Institute for Data Science in Health and Medicine, Xiamen University, Xiamen 361102, China; 3Department of Oncology, Zhongshan Hospital of Xiamen University, School of Medicine, Xiamen University, Xiamen 361004, China; 24520221154629@stu.xmu.edu.cn (Y.H.); 24520231154759@stu.xmu.edu.cn (F.L.); 24620192204532@stu.xmu.edu.cn (X.H.); 4Shenzhen Research Institute of Xiamen University, Shenzhen 518057, China

**Keywords:** pancreatic cancer, exocyst, Exo70, exosome, metastasis

## Abstract

**Simple Summary:**

In this study, we found that Exo70 promoted Pancreatic Cancer (PC) metastasis by regulating the secretion of tumor exosomes. Exo70 not only regulated the intracellular trafficking and docking of exosomes but also fused into the exosomes to promote the immune escape of PC cells. In vivo, knockdown of Exo70 or treatment with ES2 both decreased the tumor metastasis of PC cells in mice. This study provides new insight into the mechanism of invasion and metastasis in PC and identifies Exo70 as a potential prognostic factor and therapeutic target for PC.

**Abstract:**

Pancreatic cancer (PC) is an aggressive and fatal malignant tumor, and exosomes have been reported to be closely related to PC invasion and metastasis. Here we found that Exo70, a key subunit of the exocyst complex, promoted PC metastasis by regulating the secretion of tumor exosomes. Clinical sample studies showed that Exo70 was highly expressed in PC and negatively correlated with patients’ survival. Exo70 promoted PC cell lines’ invasion and migration. Interestingly, knockdown of Exo70, or using an Exo70 inhibitor (ES2) inhibited the secretion of tumor exosomes and increased the accumulation of cellular vesicles. Furthermore, Exo70 was found to accumulate in the exosomes, which then fused with neighboring PC cells and promoted their invasion. Moreover, Exo70 increased the expression of exosomal PD-L1, leading to the immune escape of PC cells. In vivo, knockdown of Exo70 or treatment with ES2 both decreased the tumor metastasis of PC cells in mice. This study provides new insight into the mechanism of invasion and metastasis in PC and identifies Exo70 as a potential prognostic factor and therapeutic target for PC.

## 1. Introduction

Pancreatic cancer (PC) is a deadly disease with the worst prognosis among common solid tumors [1]. With the advancement of modern medical technology, breakthroughs have been made in the treatment of many cancers. However, the effectiveness of therapeutic interventions for pancreatic cancer in terms of prevention, diagnosis, treatment and prognosis is still poor [2,3]. Therefore, identifying biomarkers and therapeutic targets involved in pancreatic cancer tumorigenesis is important to improve the overall survival of pancreatic cancer patients.

Exosomes are extracellular vesicles 30–150 nm in diameter containing proteins, lipids and genetic material, including DNA, RNA and miRNA. After exosomes are released from the cell, they perform a number of important extracellular functions, including interacting with the cellular microenvironment through morphogenetic signaling, immune mediation, cellular recruitment and horizontal transfer of genetic material [4]. In recent years, attention has been drawn to the role of exosomes in invasive metastasis of pancreatic cancer. Pancreatic cancer exosomes contribute to pre-metastatic niche formation in the liver [5,6]. The pancreatic ductal adenocarcinoma (PDAC) derived exosomal CD44v6/C1QBP complex is essential for the formation of a fibrotic liver microenvironment and PDAC liver metastasis [5]. Macrophage migration inhibitory factor (MIF) is highly expressed in PDAC-derived exosomes, and blocking MIF prevents hepatic pre-metastatic niche formation and metastasis [6]. PC cell-derived exosomal FGD5-AS1 causes M2 macrophage polarization to promote the malignant behaviors of PC cells [7]. Exosomal DNAJB11 can enhance the invasive ability of poorly invasive PC cells, and DNAJB11 promotes cancer development through the EGFR/MAPK signaling pathway [8].

The exocyst is an octameric complex of eight subunits, consisting of Sec3, Sec5, Sec6, Sec8, Sec10, Sec15, Exo70 and Exo84. It is mediated in the tethering/docking, priming and fusion of vesicles with the plasma membrane [9,10]. Exo70 is a key component of exocysts. The location and function of Exo70 in mammalian cells are characterized by a variety of features. Exo70 is mainly located in the cell membranes, cell junctions and nuclei of mammalian cells, and is involved in many physiological processes such as the maintenance of cell morphology, cell migration, cell junctions, mRNA splicing and cytotoxicity [11,12]. Exo70 also plays an important role in human cancer progression. The BRAFV600E mutation resulted in the phosphorylation of cortactin and Exo70, promoting actin reorganization and MMP efflux, which in turn led to melanoma cell invasion [13]. In our previous study, we found that Exo70 is a substrate for ULK1 and is cross regulated by Erk1/2 and ULK1 for the on/off of metastasis [14]. Our other study demonstrated that inhibition of Exo70 enhanced the killing effect of cisplatin and reversed the acquired drug resistance in ovarian cancer cells [15]. However, it remains to be seen whether Exo70 is involved in the tumorigenesis, progression, or invasion of PC.

In this work, we found that Exo70 can promote the invasion of PC cells through the regulation of exosome secretion. Moreover, Exo70 can be loaded to the exosomes, which then go into other neighboring cells to affect the tumor microenvironment. Our finding may provide a potential therapeutic target for the clinical treatment of pancreatic cancer.

## 2. Materials and Methods

### 2.1. Patient Eligibility

All clinical tissue specimens used in this study were from Zhongshan Hospital of Xiamen University. This study was conducted with the informed consent of all patients and approved by the Medical Ethics Committee of Zhongshan Hospital Affiliated to Xiamen University (xmzsyyjy-2021-151, approval date 13 September 2021). 

### 2.2. Cell Culture

Panc-1 cell lines and 293T cells were obtained from the Cell Bank/Stem Cell Bank of the Chinese Academy of Science (Shanghai, China). A818-4 cells were a gift from Professor Margot Zoller, the University of Heidelberg, Germany. All cell lines were identified by STR (Short Tandem Repeat) profiling by the source. Cells were cultured at 37 °C and 5% CO_2_ and in high-glucose DMEM (Gibco, Grand Island, NY, USA) with 10% fetal bovine serum (FBS) (HyClone, Logan, UT, USA) and 1% penicillin–streptomycin (Life Technologies, Carlsbad, CA, USA). 

### 2.3. Antibodies and Reagents

The following antibodies and reagents were used for this study: anti-β-actin (Sigma-Aldrich, St. Louis, MO, USA, cat#a3854), anti-ALIX (Abcam, Cambridge, UK, cat#ab186429), anti-CD63 (Santa Cruz Biotechnology, Inc., Santa Cruz, CA, USA, cat#sc-5275), anti-CD81 (Proteintech, Wuhan, China, cat#66866-1-lg), anti-CD9 (Proteintech, cat#60232-1-lg), anti-Tsg101 (Abcam, cat#ab133586), anti-Exo70 (Abcam, cat#ab118792), anti-F-actin (Cytoskeleton, Inc., Denver, CO, USA; cat#PHDR1), anti-PD-L1 (Cell Signaling, Danvers, MA, USA, cat# 13684), Endosidin2 (ES2) (Cayman Chemical, Ann Arbor, MI, USA, cat#21888) and PKH67 (Sigma-Aldrich, cat#MINI67).

### 2.4. Construction of Vectors and Stable Cell Lines

The synthesized shRNA was cloned into a pLKO.1 (RRID: Addgene_52920) vector. The specific oligoribonucleotide sequences used for the sh-Exo70-1 and sh-Exo70-2 were 5′-AAGATTTCATGAACGTCTA-3′ and 5′-TGCAGGAGAATGTTGAGAA-3′, respectively. For the negative control, the sequence used was 5′-TTCTCCGAACGTGTCACGT-3′. Human Exo70 was Flag tagged by PCR and subcloned into pLVX (RRID: Addgene_85140) vector. To generate the lentivirus, the pLKO.1 vector encoding shRNA or the pLVX vector encoding Exo70, along with the packaging vectors pMDL, pVSVG and pRSV-Rev (RRID: Addgene_12253), were transfected into 293T cells. Cells were infected with the collected lentivirus of 10 μg/mL polybrene (HY-112735, MedChemExpress, Monmouth Junction, NJ, USA) for 48 h. A stable cell line was established by selection with puromycin (2 μg/mL).

### 2.5. Cell Viability and Colony Formation

Cell viabilities were detected by the Cell Counting Kit-8 (CCK-8) (Beyotime, Shanghai, China). Cells were inoculated into 96-well microtiter plates at a density of 3000 cells/well. 12 h later, CCK-8 solution was added daily at a fixed time point. After 2 h of incubation at 37 °C, the absorbance at 450 nm was measured. For the colony-forming assay, 300 cells were seeded in 6-well plates. Then, the cells were cultured for two weeks. Colonies were then fixed with 4% paraformaldehyde and stained with 1% purple crystal. Clones containing cells were counted. Each result was analyzed in triplicate and counted for the full field of view.

### 2.6. Western Blot

Cells were lysed with RIPA lysis buffer containing protease inhibitors (#693132001, Thermo Fisher Scientific, Waltham, MA, USA) to extract protein. Protein concentrations were measured using a bicinchoninic acid (BCA) assay kit (#23227, Thermo Fisher Scientific). Sodium dodecyl sulfate polyacrylamide gel electrophoresis (SDS-PAGE) was then performed. Polyvinylidene fluoride (PVDF) membranes were used to transfer proteins (60–70 min) and blocked with 5% skimmed milk. The specific primary antibody (1:1000) was incubated with the PVDF membrane at 4 °C overnight. HRP-conjugated secondary antibodies (1:100,000) were then added. Protein expression was detected using the Molecular Imager^®^ Gel Doc™ XR System (Bio-Rad, Hercules, CA, USA) with the Enhanced Chemiluminescence (ECL) system (#P10300, New Cell & Molecular Biotech, Hong Kong, China).

### 2.7. Transwell Assay

First, cells (5 × 10^4^ cells/well) were plated in a serum-free medium in 8.0 μm pore size chambers (Millipore Corp., Billerica, MA, USA) and precoated with Matrigel matrix gel for invasion assay. Then, the medium containing 10% FBS was placed in the lower chamber as a chemotactic inducer. After 37 °C incubation for 18–24 h, the small chambers were removed, and any cells remained in the upper chamber were carefully wiped away with a cotton swab. The cells were fixed with 4% paraformaldehyde for 15 min, and then stained with 0.1% methanol crystal violet for 20 min. A full-field image of each chamber was captured for counting using an inverted light microscope at 4× field of view.

### 2.8. Exosomes Extraction

The medium was removed when the cell density reached 80%. Cells were washed three times with PBS, added to a serum-free medium and incubated for 48 h. The cell supernatant was then collected and centrifuged at 4 °C 300× *g* for 10 min to remove dead cells. The supernatant in a new centrifuge tube was then centrifuged at 4 °C, 2000× *g* for 15 min to remove cell debris. After another centrifuge at 10,000× *g* for 1 h at 4 °C to remove large EVs and heterogeneous proteins, the supernatant was carefully collected into a Beckman ultracentrifuge tube (adapted with a 70Ti rotor head). The precipitate after the last centrifugation for 90 min at 4 °C 120,000× *g* using a BECMAN L-100XP ultracentrifuge was resuspended by adding 50–100 μL of PBS, and exosomes were obtained.

### 2.9. Nanoparticle Tracking Analysis

Samples were diluted with a PBS buffer without any nanoparticles to a concentration of 1–20 × 10^8^ particles/mL for analysis. The number and size of exosomes were analyzed using a NanoSight NS 300 system (NanoSight Technology, Malvern, UK). Each sample was measured three times, and the number and size distribution of exosomes were recorded. 

### 2.10. Immunofluorescence

Cells were washed once with PBS, and then fixed in 4% paraformaldehyde for 15 min. Fixed cells were washed 3 times with PBS, and further permeabilized with 0.1% Triton X-100 for 10 min, then blocked with bovine serum albumin (BSA) (#A610903, Sangon Biotech, Shanghai, China) for 1 h. Incubation with a specific primary antibody (1:100) at 4 °C was performed overnight. Then, the cells were incubated with the specific primary antibody in a 4 °C humidified chamber overnight. Cells were incubated with donkey anti-rabbit IgG Alexa Fluor 594 (1:500, Thermo Fisher Scientific), donkey anti-mouse IgG Alexa Fluor 488 (1:500, Thermo-Fisher Scientific), and DAPI (Beyotime), respectively, for 2 h at room temperature in the dark, and then rinsed four times with PBS. Fluorescence images were obtained under a confocal laser scanning microscope with the 60× oil microscope (ZEISS, Oberkochen, Germany, cat#LSM880).

### 2.11. Transmission Electron Microscopy

Cells were fixed at 4 °C with 5% glutaraldehyde prepared in a 0.1 M phosphate buffer. Fixed cells were embedded in resin and then sectioned. After staining, observations were made using an electron microscope. For exosomes, a 10 μL exosome suspension was placed on the carbon film specimen (200 mesh, Beijing Zhongjingkeyi Technology Co., Ltd., Beijing, China) for electron microscopy. Negative staining was performed with 20 μL of 1% Uranyl acetate dihydrate for 15 s, and then dried naturally for 3 h. All samples were analyzed using an H-7650 electron microscope at 80 KV.

### 2.12. Fluorescent Labeling and Transfer of Exosomes

The extracted exosomes from A818-4-OE-Exo70 cells were labeled with PKH67 (Green) (Sigma-Aldrich, MINI67). Next, the fluorescent-labeled exosome was co-cultured with A818-4-WT cells for 48 h. The A818-4-WT cells were stained with 2-(4-Amidinophenyl)-6-indolecarbamidine dihydrochloride (DAPI) (Beyotime, Shanghai, China) and observed under an inverted fluorescence microscope to determine whether A818-4-WT cells could endocytose the exosome from A818-4-OE-Exo70 cells.

### 2.13. Animal Experiments

All animal experiments were conducted following protocols approved by the Animal Care and Use Committee of Xiamen University. The animal ethics number is XMULAC20220179. Female athymic nu/nu mice between 4 and 6 weeks of age were housed in individually ventilated cages on a 12 h light–dark cycle at 21 to 23 °C and 40% to 60% humidity. Mice were allowed free access to an irradiated diet and sterilized water. Under general anesthesia, A818-4 cells (3 × 10^6^ cells in 100 μL PBS) were injected into the tail of the pancreas. ES2 (6 mg/kg) was administered intraperitoneally every two days. After 24 days of treatment, mice were euthanized. To detect luciferase expression, mice were injected with luciferin at a dose of 100 mg/kg of body weight 12–15 min prior to imaging, anesthetized with isoflurane. Imaging was performed with a Caliper IVIS Lumina II Kinetic system (Caliper Life Sciences, Waltham, MA, USA), and the total photon flux was calculated by Living Image 4.4.

### 2.14. Evaluation of Immunohistochemical Staining

The immunohistochemical technique (IHC) was used to detect the expression of Exo70 in pancreatic cancer tissue samples. The IHC staining results were evaluated by two pathologists. The judgment of the results adopted a semi quantitative method. Scores were based on the proportion of positive cells and the intensity of cell staining. The proportion of positive cells counted were scored as follows: <10% (0 points), 10–25% (1 point), 25–50% (2 points), 50–75% (3 points) and >75% (4 points). The intensity of dyeing was divided into four levels, with 0 points for colorless, 1 point for light yellow, 2 points for brownish yellow and 3 points for brownish brown. The IHC staining score was obtained by multiplying the intensity score of staining with the proportion score of positive cells. The product of two types of scores up to 3 points was indicated by “+”; 4 points indicated by “++”; a score above 5 was indicated by “+++”; “+,++”, indicated low expression; and “+++” indicated high expression.

### 2.15. Statistical Analysis

Statistical analyses were performed using GraphPad Prism (Version 9.0, San Diego, CA, USA) and SPSS (Version 24.0, Chicago, IL, USA). Data were presented as means ± SEM of at least three independent experiments. OS and PFS were estimated by the Kaplan–Meier method with a log–rank analysis. *p* < 0.05 was considered to indicate statistical significance.

## 3. Results

### 3.1. Exo70 Expression in Pancreatic Cancer Tissues Correlated with the Prognosis of Pancreatic Cancer Patients

The expressions of Exo70 in pancreatic cancer tissues were detected by immunohistochemical analysis. A total of 71 surgical samples of pancreatic cancer patients (cohort 1) were obtained from the specimen bank of Zhongshan Hospital, Xiamen University. Exo70 was expressed in both the cytoplasm and nucleus of pancreatic cancer cells, with predominantly high cytoplasmic expression; while in normal pancreatic ductal epithelial cells, Exo70 was expressed in both the cytoplasm and nucleus, but with low cytoplasmic expression (Figure 1A,B). We further analyzed differences in Exo70 expression in the cytoplasm and nucleus of pancreatic cancer cells and paired tissues and found that the expression of Exo70 in both the cytoplasm and nucleus of pancreatic cancer was higher than that in normal pancreatic tissue (Figure 1C,D). To assess the sensitivity and specificity of Exo70 (expressed in the cytoplasm and nucleus) to distinguish pancreatic cancer from normal pancreatic tissue, we used the IHC staining score of Exo70 for each sample combined with the score cutoff determined by an ROC curve analysis. The AUC of Exo70 in the cytoplasm was 0.767 (*p* = 0.000), with a sensitivity of 93% and a specificity of 94.4%; the AUC of Exo70 in the nucleus was 0.715 (*p* = 0.000), with a sensitivity of 71.83% and a specificity of 54.4% (Figure 1E,F). These results suggested that cytoplasmic Exo70 had high sensitivity and specificity in distinguishing pancreatic cancer from normal pancreatic tissue. To further determine the prognostic value of Exo70 in pancreatic cancer, we performed a Kaplan–Meier analysis of Exo70 in the cytoplasm and nucleus and included 99 postoperative pancreatic cancer follow-up cases (cohort 2). Overall survival (OS) showed significantly shorter survival in patients with high cytoplasmic Exo70 expression compared to patients with low expression (log–rank, *p* = 0.001) (Figure 1G). Patients with a low expression of cytoplasmic Exo70 had an mOS (median overall survival) of 37.9 months, while high expression patients had an mOS of 19.9 months. However, compared with patients with high nuclear Exo70 expression (mOS: 22.0 months), patients with low EXO70 expression (mOS: 30.2 months) had longer survival, but this difference was not statistically significant (log–rank, *p* = 0.191) (Figure 1H). Further Cox univariate and multivariate analyses suggested that cytoplasmic Exo70 expression was an independent risk factor for pancreatic cancer (hazard ratio HR: 2.788, *p* = 0.001) (Table 1, Table 2 and Table 3). The expression of cytoplasmic Exo70 was negatively correlated with the overall survival of pancreatic cancer patients, and Exo70 might be an independent negative prognostic factor for pancreatic cancer. We also analyzed the correlation between the expression level of Exo70 and other clinicopathological parameters in pancreatic cancer. The results showed no significant correlation between the expression level of intracellular plasma Exo70 and age, gender, tumor size and T-stage, while there was a significant clinical correlation with lymphovascular invasion (LN) (yes vs. no, *p* = 0.001) and lymph node metastasis stage (N0 vs. N1–2, *p* = 0.001) (Figure 1I). The expression level of Exo70 in the nucleus was not significantly correlated with age, sex, size of the tumor, T stage, vascular invasion or N stage, but it was strongly correlated with the degree of differentiation of the tumor (high and medium differentiation vs. low differentiation, *p* = 0.01) (Figure 1J). Univariate and multivariate analysis showed that the expression of Exo70 in the nucleus and degree of differentiation of the tumor were independent prognostic factors negatively related to the OS of pancreatic cancer patients (Table 2 and Table 3). The results of clinical studies suggested that the expression of Exo70 in the cytoplasm was closely related to invasive metastasis of pancreatic cancer, whereas the expression of Exo70 in the nucleus was closely related to the degree of differentiation of the tumor.

### 3.2. Exo70 Regulated the Invasion and Migration of Pancreatic Cancer Cells

It has previously been shown that Exo70 is involved in the regulation of some malignant phenotypes in certain tumors, but its role in pancreatic cancer is not well defined [16,17,18]. More prominent among the many malignant phenotypes of tumors are tumor proliferation, metastasis and invasion. To investigate the role of Exo70 in pancreatic cancer, we constructed pancreatic cancer cell lines stably overexpressing Exo70 and stably knocking down Exo70 in A818-4 and Panc-1 cells (Appendix A). Neither overexpression of Exo70 in pancreatic cancer cells nor Exo70 knockdown in pancreatic cancer cells affected their growth (Figure 2A–G). Previous clinical studies have shown that cytoplasmic Exo70 expression is closely associated with invasive migration in pancreatic cancer. Knockdown of Exo70 in A818-4 and Panc-1 cells significantly inhibited the invasive migration ability of A818-4 and Panc-1 cells, while overexpression of Exo70 in A818-4 and Panc-1 cells significantly promoted the invasive migration ability of A818-4 and Panc-1 cells (Figure 2H–J). Therefore, our data indicated that Exo70 promoted invasive metastasis in pancreatic cancer.

### 3.3. Exo70 Affected the Trafficking of Exosomes

To further investigate the mechanism of Exo70 in promoting the invasive migration of pancreatic cancer cells, we performed a GO enrichment analysis using our transcriptome sequencing results. We found a correlation between Exo70 expression and exosomes (Figure 3A). Exosomes have been reported to play an important role in pancreatic carcinogenesis, invasion and metastasis, and they have the potential to serve as biomarkers, targets and drug carriers for the diagnosis and treatment of pancreatic cancer [19]. First, we extracted and identified exosomes from pancreatic cancer cells for subsequent experiments (Appendix A). The expressions of exosomal markers (ALIX, TSG101, CD63 and CD9) were significantly reduced after knockdown of Exo70 (Figure 3B). The results of transmission electron microscopy also suggested a decrease in the number of exosomes after knockdown of Exo70, which was consistent with exosome marker expression (Figure 3C). Interestingly, Exo70 affected the amounts of exosomes secreted by pancreatic cancer cells, but not their vesicle size (Figure 3D). Exosomes are tiny vesicles of cells that are released outside the cell by multivesicular endosomes (MVBs) via plasma membrane fusion. We observed a significant increase in MVBs in pancreatic cancer cells by transmission electron microscopy after knockdown of Exo70 or treatment with the Exo70 inhibitor ES2 (Figure 4A–D). Immunofluorescence experiments showed that CD63 was significantly aggregated in pancreatic cancer intracellularly after knockdown of Exo70 or ES2 treatment (Figure 4E,F). This suggested that knockdown of Exo70 or ES2 treatment promoted the aggregation of MVBs in the intracellular space and prevented the fusion of MVBs with lipid membranes, which in turn reduced the release of exosomes from pancreatic cancer cells. In summary, Exo70 was involved in the secretion of exosomes from pancreatic cancer cells.

### 3.4. Exo70 Fused into the Exosomes to Promote Other PC Cells’ Invasion and Affected the Tumor Immune Microenvironment

In our study, we found that the expression of Exo70 could be detected in the exosomes of pancreatic cancer cells, and when knocking down Exo70, the expression of Exo70 in exosomes was simultaneously reduced (Figure 3B). These results suggested that Exo70, as one of the exocyst members, not only affected the secretion of exosomes from pancreatic cancer cells but also was a component of pancreatic cancer exosomes. To further study the roles of exosomal Exo70, we fluorescently labeled Exo-Exo70 (exosomes secreted by pancreatic cancer cells overexpressing Exo70) using PKH67, and subsequently tracked the exosomes into recipient cells after 12 h (Figure 5A,B). After domesticating Exo70 knockdown pancreatic cancer cells with Exo-Exo70, we found increased expression of Exo70 in Exo70 knockdown pancreatic cancer cells (Figure 5C). Invadopodia are protrusions on tumor cells that contribute to tumor cell invasion and dissemination [20]. Notably, we found Exo-Exo70 promoted invadopodium formation in the recipient cells (Figure 5B). In addition, we used exosomes to domesticate wild-type pancreatic cancer cells and discovered that the migration ability of Exo-Exo70 domesticated pancreatic cancer cells was enhanced, compared to Exo-vector (exosomes secreted by empty loaded pancreatic cancer cells) domesticated cells (Figure 5D,E). The above results suggested that Exo70 could be loaded onto exosomes from mother cells and fused into recipient cells to promote the migratory ability of recipient cells.

In recent years, immune checkpoint therapy has rapidly progressed in pancreatic cancer treatment [21]. We analyzed the correlation between Exo70 and immune checkpoints using the TCGA database and found the highest correlation between Exo70 (EXOC7) and PD-L1 (CD274) with a correlation coefficient of 0.32 (Appendix A). Exosomes have an important role in the metastatic niche as well as an emerging role as potential biomarkers for immunotherapy. Exosomal PD-L1(Exo-PDL1) is a marker of adaptive immune activation [22]. We found a significant increase in PD-L1 expression in exosomes secreted by cells overexpressing Exo70 (Appendix A), which might aid tumor immune escape and thus reduce PD1 efficacy. In addition, we performed a survival analysis based on Exo70 and PD-L1 expression. The results showed that the Exo70-low and PD-L1-low groups had the best prognosis, while the Exo70-high and PD-L1-high groups had the worst prognosis (Appendix A). The above results indicated that Exo70 might affect the pancreatic cancer tumor immune microenvironment, and Exo70 might be a prognostic marker and a therapeutic target for pancreatic cancer.

### 3.5. Exo70 Promoted Tumor Metastasis In Vivo

To further clarify the effect of Exo70 on pancreatic cancer in vivo, we constructed a mouse in situ tumor model using pancreatic cancer cells A818-4. The mice were treated with an intraperitoneal injection of ES2 (Figure 6A). The body weight of the mice was regularly monitored, and ES2 did not affect the body weight of the treated mice compared to the control group (Figure 6B). Moreover, the mouse in vivo imaging experiments suggested that ES2 significantly inhibited pancreatic cancer metastasis compared to the control group (Figure 6C,D).

Furthermore, we validated the role of Exo-Exo70 on invasive metastasis of pancreatic cancer in vivo. We constructed a mouse in situ tumor model with pancreatic cancer cells (A818-4-shCtrl/A818-4-shExo70) and regularly injected Exo-Exo70 intraperitoneally (Figure 6E). The expression of Exo70 did not affect the body weight of mice in all groups. (Figure 6F). We visualized pancreatic cancer metastasis using a mouse in vivo imaging system and found that mice with Exo70 knockdown had significantly fewer metastases. Interestingly, injection of Exo70-enriched exosomes was able to partially restore the metastatic ability of pancreatic cancer cells in mice (Figure 6G–J), suggesting that Exo70 promoted tumor metastasis in vivo.

## 4. Discussion

Exo70 is reported to be an important component of the cell membrane complex and plays an important role in human cancer [13,14,15,18]. However, the role of Exo70 in pancreatic cancer has not yet been clarified. Our results demonstrated that Exo70 promoted invasive metastasis in pancreatic cancer. Knockdown of Exo70 induced aggregation of MVBs in PC cells, resulting in decreased exosome release. This finding was consistent with another study in head and neck cancer [23]. Interestingly, we found for the first time that Exo70 was one of the exosomal components. Exo70 could be carried by exosomes into the recipient cells, which in turn exerted its function of promoting invasive metastasis.

Exosomes with diameters ranging from 30 to 150 nm are mainly composed of lipid bilayer membrane vesicles actively secreted by cells and continuously flowing in body fluids [24]. Exosomes are now considered important participants in intercellular communication, although initially underestimated as carriers of cellular waste [25]. There is mounting evidence indicating that exosomes are rich in various biologically active molecules, including nucleic acids, proteins and lipids, which can be transferred to recipient cells from donor ones, thereby implementing intracellular information transfer [26]. Moreover, exosomes can alter the tumor immune microenvironment, aiding/assisting tumor cell escape from immune surveillance [4]. In exosomes derived from gastric cancer, the NF-κB pathway guarantees the inflammatory state in the TME, and then accelerates progression of gastric cancer [27]. Besides, exosomes from pancreatic cancer mediate microphages’ M2 polarization, leading to immunodeficiency [28]. PD-L1 on the surface of tumor cells can be upregulated by IFN-γ secreted by activated T cells, and can then bind to PD-1 of T cells’ surface to inactivate them [29,30,31]. It has been shown in several studies that Exo-PDL1 will help tumor immune escape [22,32]. Exo-PD-L1 is a major regulator of tumor progression through its ability to suppress T cell activation in draining lymph nodes and its inhibition can lead to long-lasting, systemic anti-tumor immunity [33]. We also found that exosomes secreted by PC cells overexpressing Exo70 (Exo-Exo70) increased the expression of PD-L1 in exosomes (Appendix A). Although the MVBs’ intracellular trafficking has been extensively studied, the correlation between the exocyst and MVBs’ trafficking remains elusive. A study showed that RAL-1 is involved in both MVB formation and their fusion with the plasma membrane; but these functions do not involve the exocyst, a common Ral guanosine triphosphatase (GTPase) effector [34]. However, another study suggested that the PI(3)P to PI(4)P conversion on MVEs and the recruitment of the exocyst direct the exocytic trafficking of MVEs for exosome secretion [35]. Also, in head and neck cancer, it has been reported that the exocyst can regulate exosome biogenesis and participate in the malignant behavior of tumor cells [23]. In our study, we pointed out that Exo70, a member of the exocyst, played an important role in the docking of MVBs with the cell membrane. 

We also found out for the first time that Exo70 was one of the exosomal components. Because Exo70 is partially present in the inner membrane of cells, it can be easily packaged into vesicles during the formation of endocytic vesicles and finally secreted outside the cells with exosomes. This characteristic of Exo70 makes it possible to become a biomarker for liquid biopsy. At present, the limitation of tissue biopsy has been gradually recognized in the field of precision medicine. On the other hand, liquid biopsy has the advantages of minimal invasiveness, easy sample acquisition and dynamic analysis. Exosomes have been shown to circulate stably in body fluids and contain a variety of information reflecting the state of tumor progression [26,36]. The potential of exosomes as diagnostic and prognostic biomarkers has been studied in a variety of cancers. GPC-1 circulating exosomes (GPC-1 crExos) have been reported to exhibit high specificity and sensitivity (AUC = 1.0) in recognizing healthy individuals and patients with chronic pancreatitis PDAC, superior to CA199 (AUC = 0.739). In addition, the level of GPC-1 crExos correlated with tumor load and the survival of patients before and after surgery, suggesting that GPC-1 has the potential to serve as a reliable biomarker for monitoring treatment efficacy and prognosis [37]. In our study, we found that Exo70 was highly expressed in pancreatic cancer tissues and is closely related with prognosis (Figure 1). Furthermore, overexpressed Exo70 in pancreatic cancer cells resulted in an increased amount of exosomal Exo70. Exo70 crExos needs to be further analyzed in terms of the survival prognosis of PC patients. Based on our findings, we believe that Exo70 crExos might be a potential diagnostic and prognostic biomarker of liquid biopsy of PC as well as other cancers.

## 5. Conclusions

In conclusion, our results showed that Exo70 was critical for invasive metastasis of pancreatic cancer (Figure 7). Exo70 not only regulated the secretion of exosomes but also fused into exosomes to regulate the tumor microenvironment. Knockdown of Exo70 blocked MVBs intracellularly and reduced the release of exosomes, thereby inhibiting the tumor exosome pathway-dependent immune escape. This study provides new insight into the mechanism of invasion and metastasis in PC and identifies Exo70 as a potential prognostic factor for PC. Exo70 targeted therapy might improve the efficacy of PD-1 immunotherapy and provide a new strategy for PC treatment. 

## Figures and Tables

**Figure 1 cancers-16-00336-f001:**
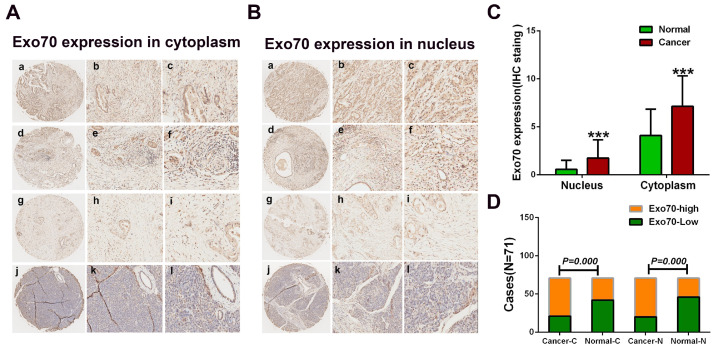
Expression of Exo70 in pancreatic cancer tissues, correlation with PC patient prognosis. (**A**) A representative image of Exo70 expression in the cytoplasm of pancreatic cancer tissue and its paired normal tissue. (a–c) High intraplasma expression of Exo70 in pancreatic cancer tissues. (d–f) High intraplasma expression of Exo70 in pancreatic normal tissues. (g–i) Low intracellular plasma expression of Exo70 in pancreatic cancer tissues. (j–l) Low intraplasma expression of Exo70 in pancreatic normal tissues. Magnification ×40 (a, d, g, and j), ×200 (b, e, h, and k), and ×400 (c, f, i, and l). (**B**) A representative image of Exo70 expression in the nucleus of pancreatic cancer tissue and its paired normal tissue. (a–c) High expression of Exo70 in the nucleus in pancreatic cancer tissues; (d–f) high expression of Exo70 in the nucleus in pancreatic normal tissues; (g–i) low expression of Exo70 in the nucleus in pancreatic cancer tissues; (j–l) low expression of Exo70 in the nucleus in pancreatic normal tissues. Magnification ×40 (a, d, g, and j), ×200 (b, e, h, and k), and ×400 (c, f, i, and l). (**C**) Immunohistochemical scores showed that the expression of Exo70 was significantly higher in both the cytoplasm and nucleus in pancreatic cancer tissues than in normal tissues (*p* = 0.000). (**D**) Comparative analysis of the ratio of high and low cytoplasmic and nuclear Exo70 expression in pancreatic cancer and normal tissues (*p* = 0.000). (**E**) ROC curves plotted by sensitivity and specificity of immunohistochemical staining for cytoplasmic Exo70 (*p* = 0.000). (**F**) ROC curves plotted by sensitivity and specificity of nuclear Exo70 immunohistochemical staining (*p* = 0.000). (**G**) Kaplan–Meier plot depicting the correlation between overall survival OS and cytoplasmic Exo70 staining score in pancreatic cancer patients (*p* = 0.000). (**H**) Kaplan–Meier plot depicting the correlation of overall survival OS with cytosolic Exo70 staining score in pancreatic cancer patients (*p* = 0.171). (**I**) Correlation analysis of the expression of Exo70 within the cytoplasm of pancreatic cancer tissues with the stage of neurovascular invasion as well as lymph node metastasis (*p* = 0.001). (**J**) Correlation analysis between the expression of Exo70 in the nucleus of pancreatic cancer tissue cells and the degree of tumor differentiation (*p* = 0.01). *** *p* < 0.001.

**Figure 2 cancers-16-00336-f002:**
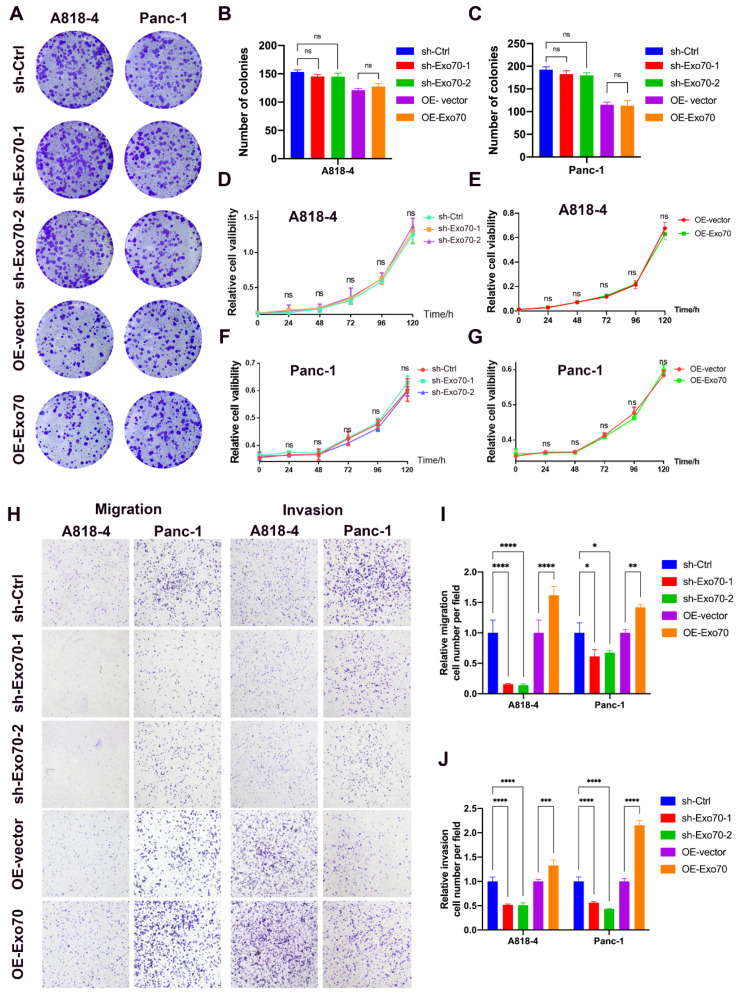
Effect of Exo70 on the proliferation and migration of pancreatic cancer cells. (**A**–**C**) Plate clone formation assay was performed to detect the effect of knockdown of Exo70 expression in A818-4 and PANC-1 cells on the proliferation of pancreatic cancer cells, ns: no statistical significance. (**D**,**F**) CCK8 assay was performed to detect the effect of Exo70 knockdown in A818-4 and PANC-1 cells on the proliferation of pancreatic cancer cells. (**E**,**G**) CCK8 assay was performed to detect the effect of Exo70 overexpression in A818-4 and PANC-1 cells on the proliferation of pancreatic cancer cells. (**H**–**J**) Transwell migration and invasion assay was performed to detect the effect of knockdown of Exo70 expression in A818-4 and PANC-1 cells on the migration and invasion ability of pancreatic cancer cells. ns *p* > 0.05, * *p* < 0.05, ** *p* < 0.01, *** *p* < 0.001, **** *p* < 0.0001.

**Figure 3 cancers-16-00336-f003:**
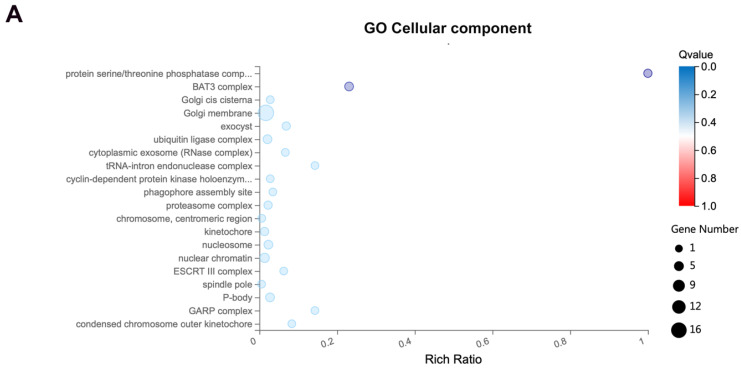
A818-4 cells inhibited exosome secretion after knockdown of Exo70. (**A**) GO enrichment analysis of differentially expressed genes between vector and OE-Exo70 A818-4 cells. (**B**) Western blot assay to analyze the expression of exosome marker protein and Exo70 in exosomes and cells after knockdown of Exo70. (**C**) Morphology and counting of exosomes after knockdown of Exo70 by transmission electron microscopy. Scale bars = 200 nm (**D**) NTA quantification of particle size of exosomes after knockdown of Exo70. ** *p* < 0.01. The uncropped bolts are shown in Appendix A.

**Figure 4 cancers-16-00336-f004:**
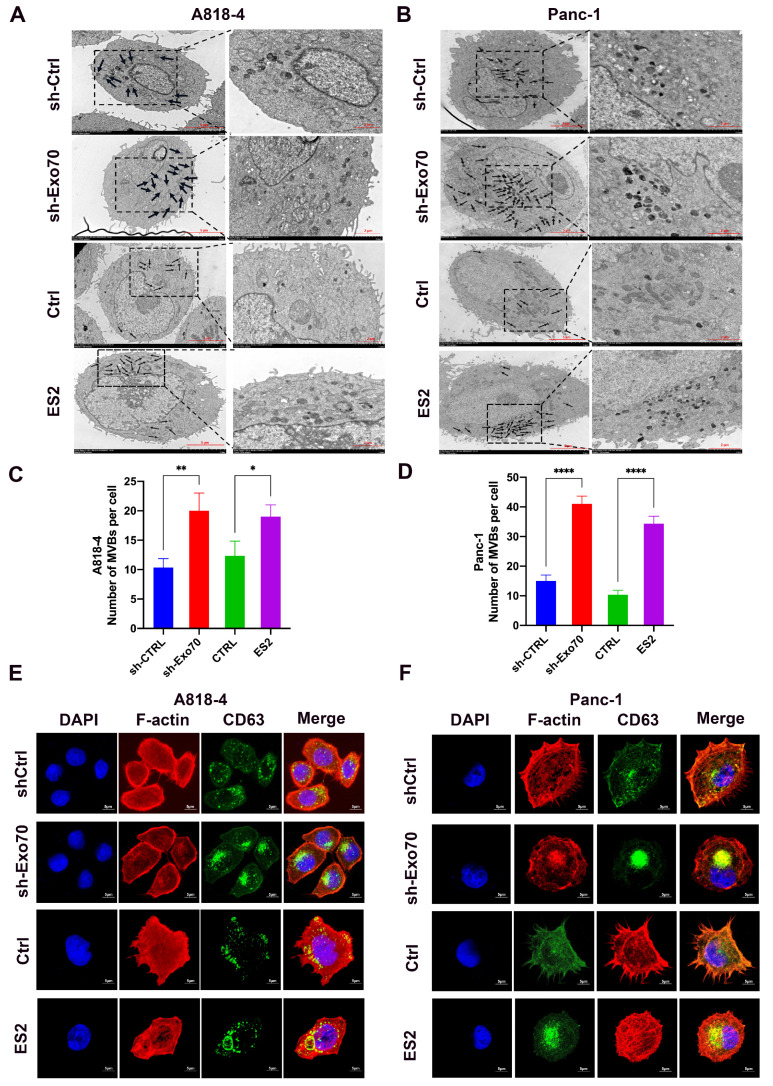
Knockdown of Exo70 caused intracellular aggregation of MVBs. (**A**,**B**) Transmission electron microscopy images of MVEs in PC cells (A818-4 and PANC-1) stably expressing control or Exo70 knockdown or in cells treated with DMSO or ES2. Scale bars = 5 μm. Black arrows indicate MVEs. Dashed rectangles indicate regions zoomed in the respective lower panels. (**C**,**D**) Quantification of the number of MVEs and the ILV numbers of each MVE in PC cells (A818-4 and PANC-1). (**E**,**F**) Immunofluorescence staining of CD63 in PC cells (A818-4 and PANC-1) stably expressing the control or Exo70knockdown. Scale bars = 5 μm. * *p* < 0.05, ** *p* < 0.01, **** *p* < 0.0001.

**Figure 5 cancers-16-00336-f005:**
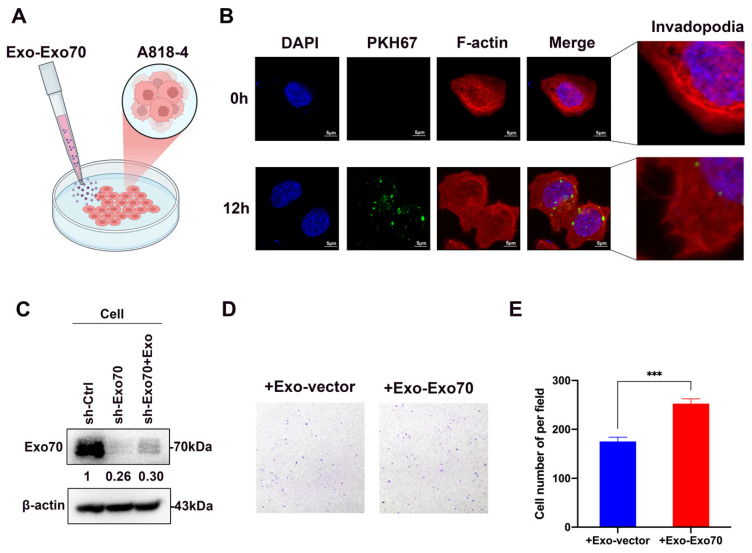
Exo-Exo70 promoted the migratory ability of recipient cells. (**A**) Schematic representation of wild-type PC cells domesticated with PKH67 fluorescently labeled Exo-Exo70. (**B**) Immunofluorescence staining of wild-type PC cells domesticated with PKH67 fluorescently labeled Exo-Exo70. Scale bars = 5 μm. Dashed rectangles indicate regions zoomed in the respective lower panels (**C**) Western blot assay to analyze the expression of Exo70 after Exo-Exo70 domestication of Exo70 knockdown PC cells. (**D**,**E**) Pancreatic cancer cells A818-4 were treated with Exo-vector and Exo-Exo70 treatments for 48 h. Transwell cell migration assay and statistical difference. *** *p* < 0.001. The uncropped bolts are shown in Appendix A.

**Figure 6 cancers-16-00336-f006:**
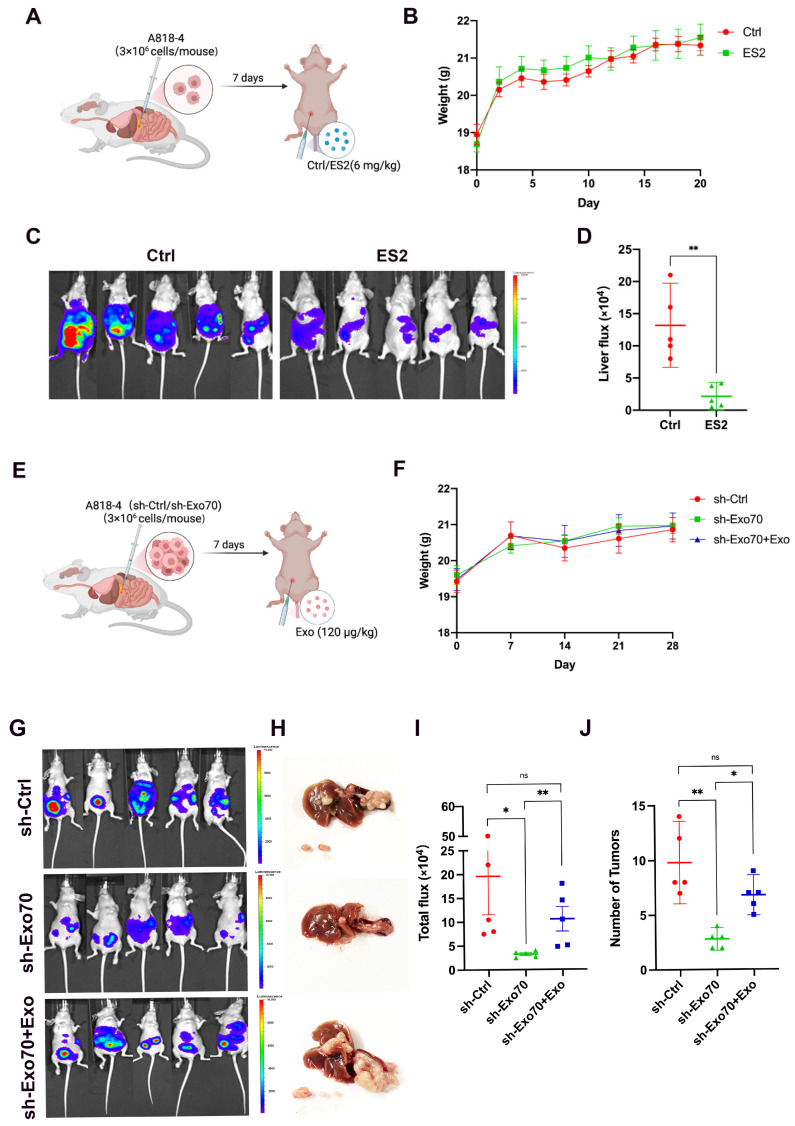
Exo70 promoted tumor metastasis in mice. (**A**,**E**) Flow diagram of the mouse experiments. (**B**,**F**) Mouse weights. (**C**) Bioluminescent imaging for A818-4-luc orthotopic xenograft colon tumors in the control and ES2 group (n = 5 per group). (**D**) Comparison of the liver bioluminescence between the control and ES2 group. (**G**) Bioluminescent imaging for A818-4-luc tumors in the sh-Ctrl, sh-Exo70 and sh-Exo70 + Exo groups (n = 5 per group). (**H**) Images of mouse livers and metastatic tumors. (**I**) Comparison of the bioluminescence between the sh-Ctrl, sh-Exo70 and sh-Exo70 + Exo groups. (**J**) Tumor number statistics. ns *p* > 0.05, * *p* < 0.05, ** *p* < 0.01.

**Figure 7 cancers-16-00336-f007:**
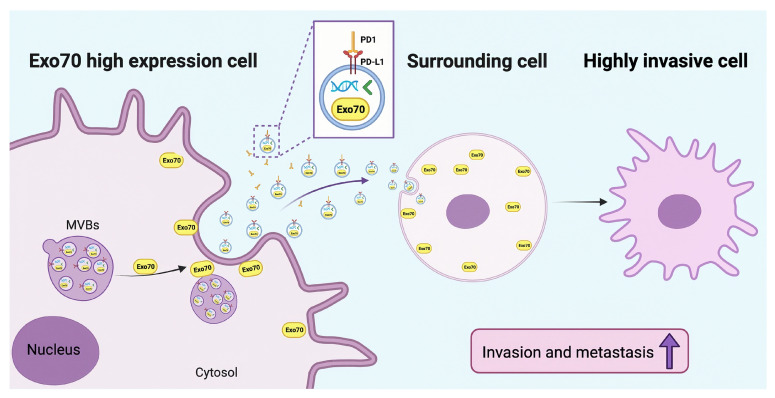
Exo70 is critical for invasive metastasis of pancreatic cancer. Exo70 not only regulates the secretion of exosomes but also fuses with exosomes into surrounding PC cells and turns them into highly invasive cells. Exo70 increases the expression of exosomal PD-L1, leading to the immune escape of PC cells.

**Table 1 cancers-16-00336-t001:** Correlation between Exo70 expression and clinicopathological features of pancreatic cancer patients (n = 99).

Characteristic	N	Exo70 Expression in the Nucleus [n (%)]		*p*	Exo70 Expression in the Cytoplasm [n (%)]		*p*
		Low	High		Low	High	
**Sex**							
**Men**	63	32	31	0.681	18	45	0.933
**Women**	36	20	16		10	26	
**Age, years**							
**≤61**	51	25	26	0.548	14	37	0.851
**>61**	48	27	21		14	34	
**Degree of tumor differentiation**							
**High/Medium grade**	67	29	38	0.01 ^a^	19	48	0.981
**Low grade**	32	9	23		11	21	
**Lymphovascular invasion**							
**No**	60	30	30	0.545	31	29	0.001 ^a^
**Yes**	39	22	17		7	32	
**T stage**							
**T1/2**	78	40	38	0.806	20	58	0.283
**T3/4**	21	12	9		8	13	
**N stage (Lymph node metastasis)**							
**0**	47	25	22	0.530	21	26	0.001 ^a^
**1 + 2**	52	27	25		7	45	

^a^ Statistically significant difference.

**Table 2 cancers-16-00336-t002:** Univariate analysis of the influence of various parameters on overall survival in pancreatic cancer patients.

Variable	OS		
	Hazard Ratio	95% CI	*p*
**Age, years**			
**≤61 vs. >61**	1.379	0.559–1.417	0.624
**Sex**			
**Men vs. Women**	1.156	0.719–1.881	0.560
**Degree of tumor differentiation**			
**High/Medium vs. Low**	1.628	1.010–2.622	0.045 ^a^
**T stage**			
**T1–T2 vs. T3–T4**	0.984	0.556–1.742	0.957
**N stage**			
**N0 vs. N1–2**	1.056	0.666–1.673	0.817
**Lymphovascular invasion**			
**Yes vs. No**	1.178	0.737–1.883	0.494
**Exo70 in the nucleus**			
**Low vs. High**	2.343	1.355–4.052	0.001 ^a^
**Exo70 in the cytoplasm**			
**Low vs. High**	1.379	0.859–2.214	0.181

Abbreviations: OS, Overall survival; CI, confidence interval; ^a^ Statistically significant difference.

**Table 3 cancers-16-00336-t003:** Multivariate analysis of the influence of various parameters on overall survival in pancreatic cancer patients.

Variable	OS		
	Hazard Ratio	95% CI	*p*
**Degree of tumor differentiation**			
**High/Medium vs. Low**	2.057	1.249–3.386	0.005 ^a^
**T stage**			
**T1–T2 vs. T3–T4**	0.980	0.543–1.771	0.948
**N stage**			
**N0 vs. N1–2**	1.031	0.647–1.644	0.897
**Lymphovascular invasion**			
**Yes vs. No**	1.315	0.805–2.148	0.274
**Exo70 in the nucleus**			
**Low vs. High**	2.788	1.539–5.048	0.001 ^a^
**Exo70 in the cytoplasm**			
**Low vs. High**	1.068	0.639–1.787	0.801

Abbreviations: OS, Overall survival; CI, confidence interval; ^a^ Statistically significant difference.

## Data Availability

All the data supporting the findings of this study are available within the article and its Appendix A and from the corresponding author upon reasonable request.

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
