# Peer review of "Exo70 Promotes the Invasion of Pancreatic Cancer Cells via the Regulation of Exosomes"

_cancers, 2024, doi:10.3390/cancers16020336_

Round 1
Reviewer 1 Report
Comments and Suggestions for Authors
This study aims to investigate the role of Exo70 in regulating PC exosomal release leading to enhanced tumorigenesis. Although Exo70 has been linked to exosmes secretion by several studies and the role tumor exosomes play in enhancing tumor invasion and metastasis, this study focuses on pancreatic cancer. However, some issues need to be solved as the following:
(1) Exo70 regulates tumor exosome secretion leading to enhanced tumor invasion, migration, and metastasis. Thus, the overexpression of Exo70 increases the boundary of released exosomes. The question is what is the impact of Exo70 on tumor exosomes tumorigenicity? If Exo70 only facilitates exosomal release, then it has no role in the tumorigenesis. The effect is coming from exosome cargo, why you didn't analyze exosomal cargo in Exo70 sh-RNA knockdown with Exo70 normal levels? You need to do RNA-Seq for exosomes from both sources to validate levels of oncogenes in both types of exosomes and conclude that Exo70 plays a role in PC tumorigenesis. Otherwise, you didn't provide real evidence of your study conclusion.
(2) The expression of PD-L1 on tumor-secreted exosomes has been reported previously but have you checked the level of PD-L1 in exosomes isolated from Exo70 KD cells compared to overexpressing cells? Using a database to address this point in your study makes no sense because PD-L1 expression in exosomes isn't your specific aim and it's not new information. Unless you would like to connect the expression of PD-L1 on exosomes with the expression of Exo70 in tumor cells. You need to provide experimental results.
(3) In Fig1, you presented the expression of Exo70 in tumor tissues cytoplasmic and nucleus expression. What is the difference between A and B? They are the same you can notice the expression in cytoplasm and nucleus in the same slide because you didn't perform a specific nucleus staining you did the same. So, B is a repeat figure. indeed, where is the normal control in IHC staining? And why do you focus on the expression of Exo70 in the nucleus and cytoplasm? what is the purpose of this discussion? Exosome biogenesis is localized in the cytoplasm not in the nucleus, I think you don't need to discuss this expression location because it means nothing in this study.
(4) In Fig2, I see no significant differences in the number of colonies (B), and tumor cell viability (F-G) in the case of Exo70-KD compared to vector only. If Exo70 has a link with tumor growth, I think we should see some differences. Can elaborate on this issue.
(5) Regarding section 3.3 page 10 line 296: as known, exosomes fusion is mediated by some signaling pathways like ESCRT pathway or ALIX pathway, have you detected any changes in these pathways in the case of Exo70-KD or not? The observation of increased multivesicular endosomes doesn’t mean exosomes are trafficking intracellularly and didn’t get released and this observation is not mechanistic evidence. You have to check the signaling of pathways regulating exosome release and then come up with a conclusion about the connection between Exo70 and exosomes release in pancreatic cancer.
Comments on the Quality of English LanguageNot bad.
Reviewer 2 Report
Comments and Suggestions for Authors
The Authors present very interesting work on the importance of Exo70 in the pathology of pancreatic cancer. The results indicate a very important role of Exo70 in invasiveness and metastatic potential of pancreatic cancer and their conclusions are backed up by a considerable amount of sound results. I strongly suggest the MS to be published as is.
Author Response
We appreciate the reviewer’s time and effort in evaluating our manuscript.
Reviewer 3 Report
Comments and Suggestions for Authors
Materials and Methods- In general, more information needs to be provided for the respective Materials/Methods/Samples
2.1 Patient samples- explain if these were FFPE samples of tumor vs matched non-tumor samples or were compared to non-tumor areas of the sections. What demographic and clinicopathological data was collected? How were tumors staged? Was follow-up data available; and how was overall survival calculated- Kaplan Meier is mentioned very briefly under Statistics, but needs some further explanation. Multivariate analysis should also be included and its use explained under Statistics.
2.2- Cell culture- what passage number/s was used for the research study?
Provide supplier information for culture medium, FBS, amount of antibiotics used and source.
2.4- There is no information on the transfection reagent used.
293T cells were used in the transfection process- provide culture conditions for 293T cells under section 2.2.
2.5- What was the purpose of using cisplatin- not explained in Methods. What was the source of the drug? Was cell staining done with crystal violet- clarify in the text? Explain how were cells counted and how many colonies/fields of view. How many biological repeats were carried out?
2.6- Provide information on kit/reagent suppliers. Was total protein extracted; and were protease inhibitors used in the extraction buffer? What percentage SDS was used for gel separation. How long was the transfer process onto PVDF membrane? Provide concentrations of primary and secondary antibodies and information on the loading control.
2.7 It would be more correct to say that -Cells that remained in the upper chamber were removed with a cotton swab (rather than "uncrossed cells").
Were the cells not fixed with methanol (and not formaldehyde) before staining with crystal violet? Please clarify.
Was an inverted microscope used and at which magnification? How many fields of view were selected for counting of cells? Were biological repeats carried out?
2.10- which objective was used to view the cells?
2.11- two processes are described under TEM- (1) cell preparation and sectioning and
(2) exosome preparation for TEM viewing on copper grids.
It is not clear to me how the fixed cells were prepared for thin slices (how were these slices done?) and then for ultra frozen sectioning - the steps for the sectioning processes are not explained. Were ultrathin sections picked up on copper grids and stained for TEM viewing? Please explain.
The exosome preparation should be separated from the cell preparation. What volume of drop was placed on the copper grid/was this air-dried/when was it stained? Which company supplied the copper grid and what are the specifications for this grid?
Results- Figure 4 A and B- the electron micrographs need scale bars on the images, or the original magnifications should be stated. Similarly with Figs 4E and F- add scale bars or include the original magnification.
Figure 5B requires magnifications.
Comments on the Quality of English LanguageThe paper needs to be reviewed by an English language editor as there are minor grammatical errors.
Reviewer 4 Report
Comments and Suggestions for Authors
This article entitled “Exo70 Promotes the Invasion of Pancreatic Cancer Cells via the 2 Regulation of Exosomes” was aimed evaluate the role of Exo70 played in the progress of metastasis and invasion of pancreatic cancer.
My comments were listed as below.
1.This article was well-organized.
2.In Line 80, there were a total of 71 clinical samples used in this study was described. However, the number of patients was 99 shown in table 1.
3.There was no definition of high or low expression of Exo70 in pancreatic cancer tissue of this study detected by immunohistochemical analysis (Line 196).
4. In figure7, why did authors use the terms of “donor cells” and “recipient cells” instead of pancreatic cancer cells? It let readers puzzled.
